# What Motivates Greenhouse Vegetable Farmers to Adapt Organic-Substitute-Chemical-Fertilizer (OSCF)? An Empirical Study from Shandong, China

**DOI:** 10.3390/ijerph20021146

**Published:** 2023-01-09

**Authors:** Xiaoyan Yi, Qinqi Zou, Zewei Zhang, Sheng-Han-Erin Chang

**Affiliations:** 1Institute of Agricultural Resources and Regional Planning, Chinese Academy of Agricultural Sciences, Beijing 100081, China; 2Key Laboratory of Arable Land Quality Monitoring and Evaluation, Ministry of Agriculture and Rural Affairs, Beijing 100081, China; 3Agricultural Production and Resource Economics, Technical University of Munich, Alte Akademie 14, 85354 Freising, Germany

**Keywords:** Organic-Substitute-Chemical-Fertilizer (OSCF), adoption behaviour, greenhouse vegetable, policy measures, psychological cognition

## Abstract

This paper reports on a study of the determinants of the adoption behaviour related to Organic-Substitute-Chemical-Fertilizer (OSCF) against the background of Green and Low-carbon Circular Agriculture (GLCA) by analysing a survey of 318 greenhouse vegetable farmers in Shandong Province, China. We use regression analyses to identify policy measures and farmers’ psychological cognition of the determinants of adoption behaviour on farmers’ psychological cognition. We use three indices for farmers’ cognition, including economic value, resource capacity, and ecosystem impact, to examine the differences between training and subsidy. Our findings showed that two policy measures (training and subsidy) had a significant positive impact on vegetable farmers’ fertilizer application. Farmers’ cognition played a mediating role. We identified and discussed the influence of policy measures on farmers’ behaviour and the mediating role of farmers’ cognition. Hence, we suggest that local governments should strengthen farmers’ training in relation to fertilizer application techniques and enhance farmers’ cognition of organic fertilizer as a substitute for chemical fertilizer in terms of economic, resource and environment aspects.

## 1. Introduction

Replacing chemical fertilizer with an organic fertilizer in greenhouse vegetable production is an important environmental management measure to improve agricultural production systems [1,2,3]. The excessive use of chemical fertilizer not only causes soil compaction, salinization and other resource and environmental problems but also increases greenhouse gas N_2_O emissions [2,4,5,6]. N_2_O is an important greenhouse gas caused by human activities, and its global warming effect is 298 times that of CO_2_ [7]. Using nitrogen fertilizer in agricultural production systems is considered to be an important source of the N_2_O emissions found in agricultural soil [8]. Promoting OSCF meets not only the need to reduce carbon emissions but also is a vital measure for making full use of livestock and poultry manure in order to encourage recycling agriculture, contributing to food security and achieving high-quality agriculture development.

In 2017, the Ministry of Agriculture and Rural Affairs (MARA) of China began to implement the policy of Organic-Substitute-Chemical-Fertilizer (OSCF) for fruit, Vegetables and tea. One hundred counties were chosen for the first batch of the pilot policy in China, of which 22 are vegetable pilot counties. Training and organic fertilizer subsidies are two main measures. Agricultural technology extension departments provide training on the role of organic fertilizer in soil and composting technology and provide technical guidance in the field to large-scale growers, cooperatives and other agricultural operators. Each county receives a national fund subsidy of 10 million yuan, and a certain percentage of subsidy funding is provided by the local county government.

In this paper, we hypothesise that when making behavioural decisions, farmers are rational and in line with the principle of optimisation. In a market economy, farmers’ decisions to use fertilizer are characterised not only by their concern about the returns on different technologies but also by their risk-averse nature in that they are more likely to adopt easily available fertilizer in the absence of policy incentives. At the same time, farmers are social people and may be influenced by external policies. Policy intervention is an effective means to solve the internalisation of agricultural production technology externalities, which can change the constraints of farmers’ decision-making and is a strong determinant of farmers’ adoption of new technologies. So, how do policy measures relating to training and subsidy influence farmers’ behaviour? What role does farmers’ psychological cognition play in policy measures and farmers’ behaviour? This paper applies the Theory of Planned Behaviour (TPB) to explain the effects of two policy measures on farmers’ behaviour and assesses the mediating effect of farmers’ psychological cognition between the two measures and OSCF technology adoption to reveal the external and internal causes of farmers’ behaviour.

The influence of farmers’ cognition on the effectiveness of environmentally friendly agriculture policy implementation has received wide attention. Some studies have concluded that the cognition that organic fertilizer substitutes for chemical fertilizer can improve the prices of apples and the cognition about the ease of organic fertilizer has significant positive effects on fruit farmers’ organic-fertilizer-use behaviour [9]. Moreover, some studies have confirmed that farmers have a low cognition of the environmental pollution caused by chemical fertilizers, insufficient knowledge regarding fertilizer application and reduction policies and a low willingness to reduce application [10]. Farmers’ environmentally friendly cognition regarding organic fertilizer cannot promote farmers’ organic fertilizer use [11]. However, some studies have verified that ecological cognition partially mediated the relationship between social norms and farmers’ willingness to recycle mulch by using ecological cognition as a mediating variable [12].

This paper aims to contribute to the existing literature by further validating the role of farmers’ cognition in agri-environmental policies and farmers’ behaviour and clarifying the variability of farmers’ cognition and the path of the mediating effects. First, we construct a systematic research framework based on theories from economics, sociology and psychology and analyse the external policy environment with policy measures and the interactions between farmers’ cognition and behaviour, which forms a more logical framework. Second, we analyse the interaction among agricultural, environmental policies, farmers’ cognition and behaviour and consider the mediating effects on farmers’ awareness of the economic value, resource capacity and ecosystem impact. Finally, we consider the path of the mediating effects of farmers’ cognition, explain each mediating effect and examine the role of farmers’ cognition in the share of these mediating effects between agri-environmental policies and farmers’ behaviour. Moreover, we measure the share of mediating effects in farmers’ cognition under two policy measures, including training and organic fertilizer subsidy.

The paper comprises five sections. Section 2 establishes the theoretical framework. Section 3 describes the research data collection, variable selection of farmers’ and the modelling methods. The results are reported in Section 4. Finally, Section 5 provides the conclusions and the discussion of policy recommendations.

## 2. Determinants of Vegetable Farmers’ Behaviour Regarding Crop Fertilization

According to Schultz’s assumption [13], small farmers will choose the means of production that can maximise their profits to achieve high agricultural productivity based on their own endowments and cost constraints. Meanwhile, the theory (CBT) emphasises the importance of cognition in problem-solving processes and the interaction between intrinsic cognition and extrinsic environment [14]. It is believed that both external behavioural and internal cognitive changes will eventually alter individual behaviour, and thereby, the behaviour will also affect cognition [15]. Therefore, we should consider starting by changing cognition to change human behaviour [16,17,18,19]. Wang et al. [20] pointed out that relevant theories in the field of social psychology could be used to explain the problems that arise in agricultural and rural development, and the cognitive–behavioural approach could be used to study the cognitive and behavioural intentions of farmers in relation to agricultural waste resource utilisation. Wang et al. [20] highlighted that cognition related to policy could adjust the production or consumption behaviour of participants in economic activities [21].

In the process of promoting policies related to OSCF, the government has guided the operating agents in various ways. First, strengthening the training related to fertilizer application to guide farmers to apply fertilizer in a rational and scientific manner. Second, using the tools of the organic fertilizer subsidy policy to reduce the cost of organic fertilizer application and encourage farmers to apply more organic fertilizer. Third, establishing experimental demonstration bases to motivate neighbouring farmers to adopt new technologies and methods. Training can strengthen farmers’ cognition concerning organic fertilizer use, which affects their judgment of the expected net benefits, and the number of training sessions received has a significant positive effect on ecological production behaviour. Organic fertilizer subsidies can reduce farmers’ expected costs of organic fertilizer use and increase their expected net benefits. The policy of agricultural technology subsidy can promote farmers’ adoption behaviour of environmentally friendly and resource-saving technologies. Organic fertilizer subsidy has a significant positive effect on fruit farmers’ behaviour relating to organic fertilizer use [22], and government policies on organic fertilizer subsidy effectively motivate fruit farmers to use organic fertilizer. Therefore, two policy incentives of training and organic fertilizer subsidy are beneficial to farmers’ organic fertilizer application behaviour. Hence, we provide Hypothesis 1 as follows:

**Hypothesis** **1.**
*Training and subsidy have a significant promotion effect on OSCF adoption behaviour of vegetable farmers.*


Cognitive Behavioural Theory (CBT) states that changes in the external environment and internal cognition can eventually influence human behaviour [23]. Policy measures may influence human behaviour by changing their cognition. Therefore, cognition plays an important mediating role in influencing behaviour. As rational economic people, farmers’ cognition in relation to OSCF begins with economic value, that is, the adoption of organic fertilizer can lead to economic benefits by increasing the crop yield or product price; secondly, the cognition concerning resource capacity is that whether the adoption of OSCF can promote the sustainable production capacity of vegetable plots, which is a key consideration for farmers; thirdly, farmers realize that the excessive application of chemical fertilizer leads to environmental pollution, such as water pollution and greenhouse gas emissions. For this reason, this study focuses on the role played by farmers’ cognition of economic value, resource capacity and the ecosystem impact of OSCF and how it works.

Thus, we provide Hypothesis 2:

**Hypothesis** **2.**
*Farmers’ cognition including economic value, resource capacity, and ecosystem impact have certain mediating roles between policy measures and greenhouse vegetable farmers’ behaviour.*


In addition, farmers’ individual and family characteristics and land features will affect farmers’ behaviour regarding crop fertilization [24,25,26,27].

In this paper, we select the characteristics of the head of the household, household characteristics and land characteristics of the farm households as the control variables.

This paper formed a research framework based on the above explanation (Figure 1).

## 3. Methods and Materials

### 3.1. Data Collections

The data used in this paper were gathered from household surveys of vegetable growers in Anqiu and Qingzhou County in Weifang City, Pingyuan, and Yucheng County in Dezhou City of Shandong Province in July 2019. The aims of this study are mainly based on the following considerations. China produces and consumes a huge number of vegetables. First, compared with food crops, vegetables have higher economic benefits, and the amount of chemical fertilizer applied to cash crops is generally more. Second, Shandong is the main production province of “vegetable basket” products in northern China. The area of greenhouse vegetables in Shandong province is about 14 million mu, accounting for about 1/4 of the country’s total area, and the output reaches more than 50 million tons per year. It has become the main hub of the Chinese vegetable industry, and its products are sold to major domestic vegetable markets. Third, we extracted the sample counties, which are policy support areas and general areas. Pingyuan County and Anqiu County are in the first batch of policy areas for the organic fertilizer substitution of chemical fertilizer on greenhouse vegetables and will receive corresponding policy support. For comparison, we chose two counties with more vegetable cultivation next to these two counties, Yucheng County and Qingzhou County. The sample counties are shown in Figure 2.

The local Bureau of Agriculture and Rural Affairs provided the information on the villages for a sample framework, from which we first selected those villages to implement greenhouse vegetable production. Then, we randomly selected 5 villages from each of the four counties. Overall, 20 households from each village were randomly selected according to the list provided by the village committee. Eventually, a total of 318 valid questionnaires were obtained. The questionnaire included the following: basic household operating conditions, vegetable planting area, input, output and farmers’ cognition of OSCF, etc. There are more than 10 vegetable crops, such as cucumbers, gourds, peppers and tomatoes, produced in the study areas (Figure 2). The content of the questionnaire was modified repeatedly, and the interviewers were also revised by the research group to ensure the scientific nature of the study and the reliability of the data.

### 3.2. Variable Selections

Based on the above theoretical framework, the dependent variable is whether to adopt OSCF. The assigned value is 1 for the adoption of OSCF and 0 for non-adoption. The independent variables include three types: (1) core independent variables, including technical training and financial subsidy. (2) Mediate variables, including farmers’ cognition of economic value, resource capacity and the ecosystem impact of OSCF. (3) Control variables, including the characteristics of the head of the household (e.g., age and education level), household characteristics (e.g., cooperative relationship, labour force numbers and greenhouse vegetable area) and the characteristics of the greenhouse vegetable plots (e.g., soil fertility and property rights’ characteristics). The description of these variables is provided in Table 1.

The adoption rate of OSCF reached 52%, and the farmers can obtain two types of organic fertilizer in the regions. One is commercial organic fertilizer, which is processed by organic fertilizer enterprises with lower use costs and less environmental pollution but a relatively higher price of about 2–4 yuan/kg. The other category is composting fertilizer, known as manure, which achieves resource utilisation through the composting fermentation of livestock and poultry manure. The reason for taking technical training and financial subsidies as core independent variables is that these two measures are the main ways to promote the policy of OSCF. Each county formulates corresponding promotion measures based on the number of organic fertilizer resources and the area of greenhouse vegetables. On the one hand, counties may carry out promotional measures, such as strengthening chemical fertilizer reduction publicity and training. In terms of training, 46% of the farmers receive OSCF-related training. The method of training includes intensive study courses and field training. The intensive study course mainly focuses on the theoretical study of fertilization content, and the field training mainly involves hiring experts to provide on-site guidance for practical problems. The training includes the amount, ratio and timing of fertilizer application for different vegetables. During the questionnaire survey, farmers participated in a wide range of training categories, such as vegetable storage and pesticide spraying. Therefore, they could only remember whether they had attended OSCF training and could not accurately state the number of times they had attended OSCF training. Therefore, this paper uses whether they had attended training as an explanatory variable.

On the other hand, each county supported by the policy will receive a subsidy of 10 million yuan and receive matching funds according to its financial level. The entities of implementation are organic fertilizer enterprises and cooperatives. (1) The organic fertilizer enterprises in the region collect livestock and poultry manure and produce organic fertilizer to provide to vegetable farmers. Farmers can purchase organic fertilizer at lower prices than the market prices, and organic fertilizer enterprises also can receive subsidies. (2) Cooperatives with subsidies are encouraged to carry out the composting and fermentation of livestock and poultry manure in the vicinity, and members can receive more favourable prices of organic fertilizer and technical guidance. There are two ways to send organic fertilizer subsidies. One is that farmers who make fertilizer by themselves or purchase compost in the pilot regions will receive a standard subsidy of 300 yuan per ton. The second is that farmers using commercial organic fertilizer will be subsidised with 50% of the price of the organic fertilizer. The percentage of receiving subsidies is 29%, which was low. In Pingyuan County, organic fertilizer production mainly relies on enterprises processing commercial organic fertilizer, and farmers receive preferential prices for organic fertilizer at an average price of 2000 yuan/t, while the average price of commercial organic fertilizer in Yucheng County, a neighbouring non-policy area county, is 2500 yuan/t. The price of commercial organic fertilizer in Pingyuan County is 25% better. In Anqiu County, organic fertilizer is mainly composted by cooperatives, and according to the research sample data, it can be seen that the cost of compost after the subsidy is about 200 yuan/t. Furthermore, some vegetable farmers can also receive free compost, while the average price of compost in Qingzhou City is about 440 yuan/t. After receiving the subsidies, the cost of using compost has dropped by at least 50%. In the survey, the farmers are indirect beneficiaries of the subsidy, so they only perceive the reduced price of organic fertilizer and do not know the exact amount of the subsidy. Therefore, we set it as a 0 or 1 variable, and 1 represents the farmers who received financial subsidies.

Based on CBT in Part 2, the cognition of economic value, resource capacity and ecological impact may have a mediating effect on farmers’ adoption of organic fertilizer as a substitute for chemical fertilizer. We used a 5-point scale for these three variables, with 1 indicating strongly disagree and 5 indicating strongly agree (Table 1).

The control variables were the characteristics of the head of the vegetable-growers household, household characteristics and vegetable plot characteristics, which were mainly used to see whether there were other factors influencing vegetable farmers’ behaviour toward using organic fertilizer instead of chemical fertilizer.

### 3.3. Empirical Models

#### 3.3.1. Logit Model

The options of vegetable farmers’ behaviour toward OSCF are yes or no, which is a typical dummy variable. The Logit model was used to analyse the selectivity problem by estimating the probability of farmers’ adoption of OSCF ($P\left(X\right)$) under the given characteristic conditions. Equations (1) and (2) exemplify this as follows:(1)$$P\left(X\right)=Pr(D=1|X)=\frac{exp(\beta X)}{1+exp(\beta X)}$$
(2)$$Logit\left[\pi \left(Y=1\right)\right]=\ln n[\frac{\pi \left(Y=1\right)}{1-\pi \left(Y=1\right)}]=\beta_{0}+\beta_{0}X_{1}+\beta_{2}X_{2}+\cdots +\beta_{M}X_{m}$$

This paper uses the mediation effect test model to empirically analyse the mediating effect of value identification to establish the mechanism of the effect on the policy of OSCF based on CBT. The relationship among the variables can be described by the following equations:(3)$$Logit\;y=cx+\beta_{i}controls_{i}+e_{1}$$
(4)$$Reg\;m_{j}=ax+\beta_{i}controls_{i}+e_{2}$$
(5)$$Logit\;y=c^{\prime }x+bm_{j}+\beta_{i}controls_{i}+e_{3}$$

In the above equations, y is the dependent variable, representing the farmers’ OSCF; x is the independent variable, which represents the core explanatory variable used in this paper, namely, training and subsidy policies. $m_{j}$ is the intermediary variable; c is the total effect of the independent variable x on the dependent variable y; a is the effect of the independent variable x on the intermediary variable $m_{j}$; $c^{\prime }$ is the influence of the direct effect of the independent variable x on the dependent variable y in control of the intermediary variable; b is the intervening variable, and the intermediary variable $m_{j}$ effect on the dependent variable y after controlling the effect of the intervening variable; $controls_{i}$ is the control variable representing the parameter to be estimated. The mediation effect is equal to the product of the coefficient a*b. In combination with the characteristics of the variables in this paper, the stepwise regression method is used to examine the mediation effect by establishing the logit and linear regression models to test the coefficients c,a,b. If all of the above effects are significant, there is proof of a mediating effect.

#### 3.3.2. Karlson–Holm–Breen-Method (KHB-Method)

In order to deeply analyse the mediating effect, the KHB method, developed by Karson, Holm and Breen, is applicable to multiple independent variables and multiple mediation variables. Therefore, it was adopted in this study to analyse the direct effect, mediating effect and the total effect of multiple independent variables and multiple mediating variables in the mediating effect model [28].

In the linear regression model, the direct effect and intermediary effect can be read directly through the comparison of the coefficients. However, in the Logit model, $y^{*}$ is the unobservable latent variable, $c_{D}$ represents the total effect and $c_{T}$ is the direct effect. $y^{*}$ is an unobservable dichotomous variable.
(6)$$y^{*}=\alpha_{D}+c_{D}x+\beta_{i}controls_{i}+\omega$$
(7)$$y^{*}=\alpha_{T}+c_{T}x+bm_{j}+\beta_{i}controls_{i}+\varepsilon$$

Between Equations (6) and (7), the $y^{*}$ value is $\left\{\begin{array}{c} y=1\;if\;y^{*}\geq \tau \\ y=0\;if\;y^{*}<\tau \end{array}\right.$, τ is the critical value. x is the independent variable, $m_{j}$ is the intermediary variable; the actual direct effect is $C_{T}=\frac{c_{T}}{\sigma_{T}}$, and the actual total effect is $C_{D}=\frac{c_{D}}{\sigma_{D}}$, where $\sigma_{D}$ and $\sigma_{T}$ are standard errors of Equations (6) and (7), respectively, $\sigma^{\prime }<\sigma$. Therefore, the mediating effect is $C_{D}-C_{T}=\frac{c_{D}}{\sigma_{D}}-\frac{c_{T}}{\sigma_{T}}$.

## 4. Results

The Collin command is used to successively diagnose the collinearity of the independent variables and the control variables in the model, which can avoid the multicollinearity problem caused by variable selection. The VIF values are all less than 3, indicating that the model is less likely to be affected by collinearity.

### 4.1. Impact of Two Policy Measures (Training and Subsidies) on Farmers’ OSCF Behaviour

It can be seen from Table 2 that the estimated coefficient of the variable, “whether to apply organic fertilizer or not”, is positively significant at the level of 1%. Both training and organic fertilizer subsidies have a positive impact on vegetable farmers’ OSCF behaviour. Moreover, Hypothesis 1 has been verified. The probability of adopting OSCF is 32.8%, and there are more farmers who have received training than those who have not received training. The significance test at the 1% level indicates that training has a positive impact on vegetable farmers’ adoption of OSCF, which is consistent with the research conclusion of Wang et al. [21]. In the process of promoting OSCF, technical training clearly promoted the behaviour of vegetable farmers to apply organic fertilizer. Farmers who receive policy subsidies are 36.8% more likely to adopt OSCF than those who do not receive subsidies, which passes the significance test at a 1% level. Organic fertilizer subsidy has a significant positive impact on the organic fertilizer application behaviour of vegetable farmers. In our previous research articles relating to the region, we found that the application amount of organic fertilizer in the policy areas increased while its cost decreased by 25%, mainly due to organic fertilizer subsidies [29]. The government subsidy for farmers’ application of organic fertilizer can make up for extra costs incurred by the farmers, and vegetable farmers are more likely to use OSCF.

The number of labourers, property rights of arable land and soil fertility have positive impacts on organic fertilizer application behaviour, indicating that farmers with more labourers in the family are more willing to adopt the organic fertilizer replacement behaviour and self-employed (or leasehold) arable farmers have higher probabilities of using organic fertilizer. The effect of the planting scale on organic fertilizer application behaviour is significantly negative, indicating that the larger the vegetable planting areas, the lower the tendency of farmers to apply organic fertilizer. Hence, Hypothesis 1 has been verified.

### 4.2. The Role of Farmers’ Cognition and Farmers’ OSCF Adoption Behaviour

Table 3 shows the results of the step-by-step regression. In Model 1, only the program measures are analysed in relation to the OSCF of greenhouse vegetable farmers. More than one control variable has been added to Model 2 based on Model 1. According to Equation (3), models 3, 5 and 7 gradually add the mediation variables: the cognition of economic value, ecosystem impact and resource capacity on the basis of Model 2. As can be seen from the estimation results in Table 3, when the regression analysis is conducted on the three cognitions of policy measures separately according to the control variables, policy measures have significantly positive impacts on the three cognitions. When policy measures and three cognitions are gradually added into the model, both policy measures and three cognitions have significant effects on the OSCF behaviours of greenhouse vegetable farmers. In other words, the cognition of economic value, resource capacity and ecosystem impact all play positive mediating roles toward adopting OSCF. Thus, Hypothesis 2 has been supported.

### 4.3. The Path of Farmers’ Cognition and Farmers’ OSCF Adoption Behaviour

The KHB method is used to test the mediating effect, for there are two core independent and three intermediary variables required to further explore the internal mechanism of the policy measures on the OSCF of greenhouse vegetable farmers. On the one hand, it verifies the direct, mediating and total effects of different policies and analyses the mediating effect contribution of different measures. On the other hand, the mediating effect is also verified by the role it plays in farmers’ cognition of the economic value, resource capacity and ecosystem impact.

Table 4 shows the decomposition results of the OSCF behaviour relating to greenhouse vegetable farmers by decomposing several key independent variables. The total effect of training on the adoption of OSCF by greenhouse vegetable farmers is 11.59, among which the direct effect is 4.222 and the mediating effect is 7.372, respectively. The cognition of farmers shows that training accounts for 63.58% of the effect on the adoption of OSCF. The total effect of the subsidies on greenhouse vegetable farmers’ adoption of OSCF is 12.87, the direct effect is 6.622, and the mediating effect is 6.253. Greenhouse vegetable farmers’ cognition of OSCF adoption stands at 48.57%.

We analysed the mediating variables separately to further reveal the path of the mediating effects in relation to different policy measures. The results are presented in two aspects in Table 5. The first one refers to the mediating effects of training, which are 2.862, 2.977 and 1.533, respectively, through the cognition of ecosystem impact, economic value and resource capacity. The results indicate that the mediating effect of training through the farmers’ cognition of economic value is the largest, accounting for 40.38% of the mediating effect. It means that training plays a key role in raising farmers’ economic value of OSCF behaviour. Moreover, training can improve farmers’ cognition of OSCF, and therefore farmers are more willing to adopt OSCF technology. The second aspect is the mediating effects of subsidies for organic fertilizer through the cognition of ecosystem impact, economic value and resource capacity, which are 2.627, 2.329 and 1.297, respectively. The effect of organic fertilizer subsidy on farmers’ ecosystem impact cognition accounts for 42% of the mediating effects. As a kind of ecosystem protection compensation policy, organic fertilizer subsidy makes up part of the opportunity cost of organic fertilizer loss for greenhouse vegetable farmers. The mediating effect of the ecosystem impact is greater than that of the economic impact. Because the organic fertilizer subsidy is modest and depends on the preferential purchase price, it has little impact on perceiving the economic value of the products for greenhouse vegetable farmers.

## 5. Discussion

### 5.1. Different Contributions between Training and Subsidy

The study shows that the mediating effect of farmers’ cognition in the two measures is different. Farmers’ cognition of training is 63.58%, and organic fertilizer subsidy is 48.57%, respectively. This finding is more consistent with the existing studies in which technical training has a significant effect on farmers’ behaviour. Ref. [30] found that agricultural training had a positive impact on Chinese farmers’ fertilizer management knowledge acquisition. Furthermore, ref. [31] confirmed that trained farmers obtained significantly more fertilizer management knowledge than non-trained farmers. Some studies further subdivided training into field instruction and course training and concluded that field instruction was more useful in improving farmers’ knowledge of fertilizer management and that in-class training programs were less useful [32]. However, some studies have shown that organic fertilizer subsidies reduce farmers’ production costs and increase the utility of farmers’ organic fertilizer choices [9], but their impacts vary widely across regions and crops. In saline areas, even subsidizing organic fertilizer does not necessarily help towards adoption [33]. In Nigeria, 100 kg of subsidized fertilizer supplied to a farm household reduced the probability of its participation in the commercial fertilizer market by 10–21% points, and their participation did not affect the use of organic fertilizer by the farmers [34]. When farmers can find changes in fertilization in the field, a principal factor influencing the use of organic fertilizer, a cost reduction through government subsidy policies is efficient. Organic fertilizer subsidies may reduce production costs in a short time, but farmers can obtain long-term benefits. Furthermore, the results also highlight the need for training, especially field guidance. Farmers can really master OSCF to achieve scientific fertilization and promote the quality and efficiency of agricultural production.

### 5.2. Variability in the Mediating Role of Farmers’ Cognition

This study showed that farmers’ economic value, resource capacity and ecosystem impact cognition had significant mediating effects on farmers’ behaviour. Moreover, the results in this study remain consistent with previous work [9,10,26]. However, we also found that economic value cognition mediated 40.38% of the technical training measures on farmers’ behaviour, while ecosystem impact cognition and resource capacity cognition accounted for 38.82% and 20.80%, respectively. Whereas the mediating role of farmers’ ecosystem impact cognition concerning organic fertilizer subsidy on farmers’ behaviour reached 42%, economic value cognition and resource capacity cognition accounted for 37.25% and 20.75%. The result indicated the path and variability in the mediating role of farmers’ cognition between policy measures and farmers’ behaviour. Yu et al. [26] demonstrated that farm household cognition had an impact on farm household behaviour but did not identify the pathways of the cognitive impact. Some studies used structural equation modelling and the Bootstrap method to identify the mediating effect [17] but only proved the significance of farmers’ cognition and did not measure the proportion of different cognitive mediating effects.

### 5.3. Differences in Regions

We found significant variability in the adoption of OSCF by greenhouse vegetable growers in the four counties studied. First, we selected two policy counties and two non-policy counties in the study area selection. For the non-policy areas, some training was adopted with no subsidy, and the adoption rate of OSCF among the vegetable farmers was low; the pilot counties had policy support, and the adoption rate was higher. Secondly, the two policy counties exhibited differences in organic fertilizer production methods. Pingyuan county mainly relied on agricultural enterprises to produce commercial organic fertilizer. Although they receive government subsidies, the sales price of organic fertilizer was only 20% better than the market price (Table 6). Therefore, the adoption rate of vegetable farmers did not reach a very high level. On the contrary, Anqiu County mainly adopted t cooperative composting and fermentation in the vicinity, with a lower production cost for organic fertilizer. Vegetable farmers can enjoy the lower price of organic fertilizer, which is 54.5% better than the market price (Table 6). According to the differences in regions, each area should choose appropriate ways to promote OSCF adoption.

## 6. Conclusions

Our study examines the policy measures of OSCF and farmers’ psychological cognition to study the determinants of the adoption behaviour of greenhouse vegetable farmers in Shandong, China. We measured the differences between training and subsidy using farmers’ cognition of economic value, resource capacity and ecosystem impact to identify and compare the path of the mediating roles. Our findings show that two policy measures have significant positive impacts on OSCF behaviour in vegetable farmers. Farmers’ cognition plays a significant mediating effect. Overall, our findings highlight the importance of training and subsidy in adoption behaviour and the mediating role of farmers’ cognition between adoption behaviour and policy measures.

## Figures and Tables

**Figure 1 ijerph-20-01146-f001:**
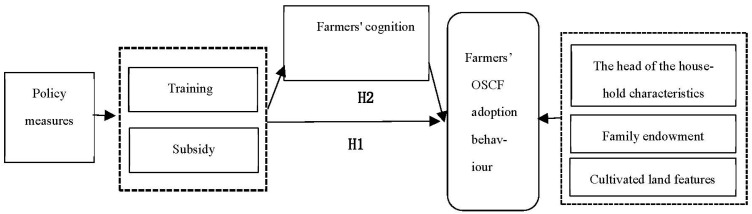
Theoretical Framework.

**Figure 2 ijerph-20-01146-f002:**
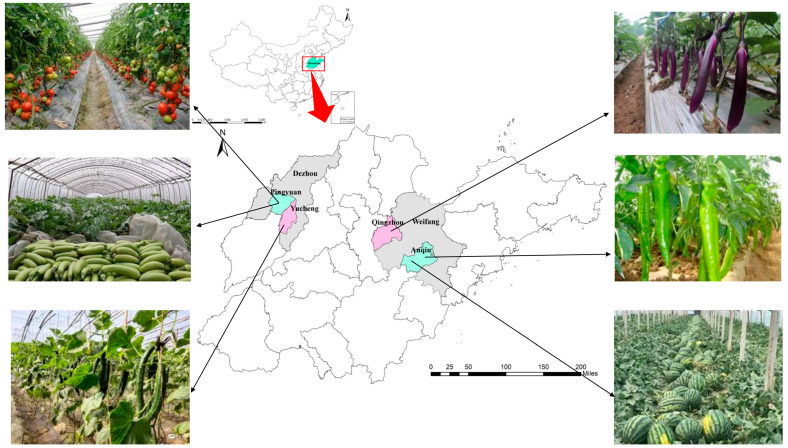
Location of study districts, Shandong, China.

**Table 1 ijerph-20-01146-t001:** Variables and descriptive statistics.

Type of Variables	Description and Assignment	Maximum	Minimum	Mean	SD
Dependent variables					
whether to adopt OSCF	1 = Yes; 0 = No	1	0	0.53	0.49
Core independent variables					
Training (X1)	1 = Yes; 0 = No	1	0	0.45	0.49
Subsidy (X2)	1 = Yes; 0 = No	1	0	0.29	0.45
Mediate variables					
Economic value (M1)	Do you think OSCF will increase vegetable income? Level 5:1 = strongly disagree, 5 = strongly agree	5	1	3.25	1.07
Resource capacity (M2)	Do you think OSCF is good for resource capacity of vegetable plots? Level 5: 1 = strongly disagree, 5 = strongly agree	5	1	3.03	1.18
Ecosystem impact (M3)	Do you think OSCF can improve agri-ecological environment? Level 5: 1 = strongly disagree, 5 = strongly agree	5	1	3.11	0.99
Control variables					
The head of the household characteristics					
Age (C1)	ordinal	83	27	52.02	8.58
Education (C2)	1 = primary school and illiterate, 2 = junior school, 3 = high school, 4 = junior college and higher education	4	1	2.00	0.65
Family characteristics					
Cooperative (C3)	Whether to join the cooperative? 1 = Yes; 0 = No	1	0	0.47	0.50
Labour force (C4)	ordinal	6	0	2.87	1.00
Greenhouse vegetable area (C5)	Ha	5.09	0.03	0.51	0.43
Characteristics of greenhouse vegetable plots (C6)	1 = barren; 2 = in general; 3 = fertile	3	1	2.53	0.54
Property rights (C7)	1 = management right, 2 = Self-contracting rights	1	0	0.49	0.50

**Table 2 ijerph-20-01146-t002:** The direct effect of program measures on farmers’ OSCF behavior.

Variables	Coef.	S.E	Marginal Effect
Training(X1)	4.873 ***	(0.651)	0.328 ***
Subsidy(X2)	5.465 ***	(0.807)	0.368 ***
Age(C1)	0.036	(0.027)	0.002
Education(C2)	−0.351	(0.384)	−0.024
Cooperative(C3)	0.303	(0.472)	0.020
Labours(C4)	1.207 **	(0.570)	0.081 **
Greenhouse vegetable area(C5)	−2.886 *	(1.478)	−0.194 **
Soil fertility(C6)	0.816 **	(0.413)	0.055 **
Property rights(C7)	1.576 ***	(0.474)	0.106 ***
Constants	−8.461 ***	(2.376)	
*N*	318		318
Wald test	90.71 ***		
pseudo *R*^2^	0.663		0.663

Note: *** denotes 1% level of significance, ** denotes 5% level of significance, * denotes 10% level of significance.

**Table 3 ijerph-20-01146-t003:** The mediating effect of farmers’ cognition.

Variables	Logit	Logit	Logit	Reg	Logit	Reg	Logit	Reg	Logit
	Y1 Model (1)	Y1 Model (2)	X,M,Y1 Model (3)	M1 Model (4)	Y1 Model (5)	M2 Model (6)	Y1 Model (7)	M3 Model (8)	Y1 Model (9)
Training	4.056 ***	4.873 ***	4.736 ***	0.918 ***	4.648 ***	0.952 ***	4.242 ***	1.038 ***	4.222 ***
	(9.56)	(7.49)	(4.91)	(9.98)	(5.18)	(8.78)	(5.78)	(8.38)	(2.98)
Subsidy	4.811 ***	5.465 ***	4.178 ***	0.842 ***	6.969 ***	0.745 ***	5.682 ***	0.878 ***	6.622 ***
	(6.18)	(6.77)	(4.69)	(8.15)	(3.38)	(6.92)	(6.28)	(6.67)	(5.58)
Economic value					3.113 ***				3.127 ***
					(3.37)				(4.22)
Ecosystem impact			3.199 ***						3.119 *
			(3.90)						(1.81)
Resource capacity							2.221 ***		1.477 **
							(5.25)		(2.32)
Control variables	Uncontroled	Controlled							
Constants	−2.356 ***	−8.461 ***	−19.66 ***	2.312 ***	−18.15 ***	1.494 ***	−16.13 ***	2.179 ***	−40.18 ***
	(−8.11)	(−3.56)	(−4.86)	(5.62)	(−3.25)	(3.18)	(−4.56)	(4.39)	(−3.56)
*N*	318	318	318	318	318	318	318	318	318
*R* ^2^				0.468		0.412		0.424	
pseudo *R*^2^	0.616	0.663	0.830		0.857		0.796		0.943

Note: *** denotes 1% level of significance, ** denotes 5% level of significance, * denotes 10% level of significance.

**Table 4 ijerph-20-01146-t004:** The decomposed mediating effect of different policy measures.

	Coef.	Compare Average Local Effects
Training		
Total effects	11.59 ***	0.125 ***
	(5.66)	(5.29)
Direct effects	4.222 ***	0.0406 *
	(2.98)	(1.87)
Mediating effects	7.372 **	0.0842
	(2.31)	
Proportion of mediating effect (%)	63.58	
Subsidy		
Total effects	12.87 ***	0.139 ***
	(4.70)	(5.38)
Direct effects	6.622 ***	0.0690 ***
	(5.58)	(2.97)
Mediating effects	6.253 **	0.0698
	(2.11)	
Proportion of mediating effect (%)	48.57	

Note: *** denotes 1% level of significance, ** denotes 5% level of significance, * denotes 10% level of significance.

**Table 5 ijerph-20-01146-t005:** The decomposed mediating effect of farmers’ cognition in different policy measures.

Variables	Coef.	S.E	Proportion of Mediating Effect (%)
Training			
Economic value	2.977	1.121	40.38
Resource capacity	1.533	0.770	20.80
Ecosystem impact	2.862	1.050	38.82
Subsidy			
Economic value	2.329	0.914	37.25
Resource capacity	1.297	0.663	20.75
Ecosystem impact	2.627	0.981	42.00

**Table 6 ijerph-20-01146-t006:** Investigated counties and vegetable farm household samples.

Region		Adoption Ratio (%)	Percentage of Farmers Receiving Subsidies (%)	Price of Organic Fertilizer (yuan/t)	Method of Getting Organic Fertilizer (1 = Commercial Organic Fertilizer, 2 = Compost)
Non-policy county	Yucheng	29.5	0	2500	1
	Qingzhou	39.2	0	440	2
Policy county	Pingyuan	51.1	50	2000	1
	Anqiu	86.7	77.7	200	2

Note: Data came from the authors’ survey.

## Data Availability

The data presented in this study are available upon request from the corresponding author.

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
