# Peer review of "What Motivates Greenhouse Vegetable Farmers to Adapt Organic-Substitute-Chemical-Fertilizer (OSCF)? An Empirical Study from Shandong, China"

_ijerph, 2023, doi:10.3390/ijerph20021146_

Round 1

Reviewer 1 Report

General comment:

How are the terms economic, environmental, and sustainable defined in this paper?

Today, sustainability generally comprises the three pillars of ecological, social, and economic. Why does the paper distinguish between sustainable and ecological?

The title says :Greenhouse vegetable farmers.

The text says almost nothing about Greenhouse vegetable farmers. In the data collection, farm size is collected, but no statement about greenhouse area. There is also no information on the ratio of greenhouse area / farm size.

Introduction

The introduction is very long and often unclear. A large amount of literature is cited, but not critically evaluated. Literature is also cited that lists influencing factors that are not examined in the author's own study. Example:

Line 81: “Farmer's operation characteristics, …  farm size play positive roles in influencing farmers' choice of organic fertilizers rather than chemical fertilizers” The opposite is stated in the present study.

Line 84. “Social norms, social trust, capital endowment, credit demand and other factors can affect farmers' behaviour concerning their use of fertilizers (Guo et al., 2020; Wei et al., 2020; Zhao and Xia, 2020).” In the present survey, these factors do not play a role.

Remark to Fg 1: “The head of the household characteristics” -> The characteristics of head of the household” should have an effect on Farmers Perception, not on OSCF

The introduction should be tightened and focus on the relevant aspects. Perhaps it is better to start with figure 1and explain the structure step by step.

3.2 Variable selections.

The empirical survey is insufficient to provide an answer to the research question.

1. It is unclear how much training the farmers received.

2. It is unclear how high the subsidies are and whether they compensate for the additional costs.

3. It is unclear in what form organic fertilizer is applied and whether it is available.

4. Why is the distinction between sustainable and envirnmental?

5 . Only school education iis asked, but not practical experience.

6. Why the distinction between sustainable and envirnmental in relation to soil?

7. Only the farm size is collected, what about the greenhouse area?

4. Results

Figure.3 left not necessary, results are already included in Tab 1.

“Table 2. Regional differences of farmers OSCF” is is interesting, but it lacks an attempt to explain the large differences between regions.

Line 293: “As rational smallholder farmers, vegetable farmers focus on maximizing their economic profits, and the cost of using organic fertilizers is higher than that of chemical fertilizers.” Yes, that is economic rationality. But there is a lack of information in the text and in the survey about the costs and government subsidies or the relationship between them.

Line 323. “Thus, hypothesis H3 has been verified” Surprisingly how can a hypothesis be verified that has not been formed (There is no H3 in your Text!)

Line 363: “As a kind of ecological protection compensation program, the organic fertilizer subsidy program makes up the opportunity cost of organic fertilizer loss for green-364 house vegetable farmers”. How can this be deduced from the empirical findings in this paper?

The paper should be more clearly focused and only those aspects should be discussed that were also empirically queried. However, it is questionable whether this captures and explains reality.

Missing references:

XIE Hai-kuan and XU Chi, 2019; 68 Yanli, 2015

Lu et 79 al., 2019

Schultz, 1964

Author Response

Dear Reviewer:

Many thanks for your detailed and valuable comments and suggestions on our manuscript entitled “What motivates greenhouse vegetable farmers to adapt Organic-Substitute-Chemical-Fertilizer(OSCF)? An empirical study from Shandong, China?” (ijerph-2117168). The comments and suggestions are very helpful for improving our paper. We have made revision based on the comments and suggestions. Please find our response as follows. Point1: How are the terms economic, environmental, and sustainable defined in this paper? Today, sustainability generally comprises the three pillars of ecological, social, and economic. Why does the paper distinguish between sustainable and ecological?

Response 1: Thank you for your suggestion. I may not have defined clearly the meaning and precise expression of core variables. After reflection, we want to observe farmers' cognition of economic value, resource capacity, and ecosystem impact of OSCF in this article. Therefore, we clearly parsed and defined the variable selection in the second and third part of paper.

In the second part, we added the path of the effect of three cognitions.

"As rational economic people, farmers' cognition of OSCF begins with economic value, that is, the adoption of organic fertilizer can bring economic benefits by increasing in crop yield or product price; secondly, the cognition of resource capacity is that whether the adoption of OSCF can promote the sustainable production capacity of vegetable plot is a key consideration for farmers; thirdly, farmers realize that excessive application of chemical fertilizer  brings environmental pollution such as water pollution and greenhouse gas emissions. For this reason, this study focuses on the role played by farmers' cognition of economic value, resource capacity, and ecosystem impact of OSCF, and how it works."

In the third part, we added the analysis of the three cognitions.

"Based on CBT in Part 2, the cognition of economic value, resource capacity and ecological impact may have a mediating effect on farmers' adoption of organic fertilizer as a substitute for chemical fertilizer. We used a 5-point scale for these three variables, with 1 indicating strongly disagree and 5 indicating strongly agree (Table 1)."

 Point2: The title says : Greenhouse vegetable farmers. The text says almost nothing about Greenhouse vegetable farmers. In the data collection, farm size is collected, but no statement about greenhouse area. There is also no information on the ratio of greenhouse area / farm size.

Response2: Thank you very much for this comment! We are sorry that this is not clear in the paper. In fact, the object of this article are greenhouse vegetable farmers from data collection to results analysis and the planting area only refers to the area of greenhouse vegetable planting without including other crops. Therefore, we have made further clarification in the paper.

 Point 3: The introduction is very long and often unclear. A large amount of literature is cited, but not critically evaluated. Literature is also cited that lists influencing factors that are not examined in the author's own study. Example: Line 81: “Farmer's operation characteristics, …  farm size play positive roles in influencing farmers' choice of organic fertilizers rather than chemical fertilizers” The opposite is stated in the present study. Line 84. “Social norms, social trust, capital endowment, credit demand and other factors can affect farmers' behaviour concerning their use of fertilizers (Guo et al., 2020; Wei et al., 2020; Zhao and Xia, 2020).” In the present survey, these factors do not play a role. Remark to Fg 1: “The head of the household characteristics” -> The characteristics of head of the household” should have an effect on Farmers Perception, not on OSCF. The introduction should be tightened and focus on the relevant aspects. Perhaps it is better to start with figure 1and explain the structure step by step.

Response 3: Thank you very much for your advice. The logic of the article introduction is very important, so we have made changes to remove some literature that is less relevant to this article and further sorted the literature around the article topic.

" Replacing chemical fertilizer with organic fertilizer in greenhouse vegetable production is an important environmental management measure to improve agricultural production systems [1-3]. Excessive use of chemical fertilizer not only causes soil compaction, salinization and other resource and environmental problems, but also increases greenhouse gas N2O emissions [2,4-6]. N2O is an important greenhouse gas caused by human activities, and its global warming effect is 298 times than that of CO2 [7]. Using nitrogen fertilizer in agricultural production systems is considered to be an important source of N2O emissions found in agricultural soil [8]. Promoting organic fertilizer instead of chemical fertilizer not only meets the need of reducing carbon emissions, but also is a vital measure of making full use of livestock and poultry manure in recycling agriculture, contributing to food security and achieving high-quality agriculture development.

In 2017, the Ministry of Agriculture and Rural Affairs (MARA) of China began to implement the policy of Organic-Substitute-Chemical-Fertilizer (OSCF) in Fruit, Vegetable and Tea. One hundred counties were chosen as the first batch of pilot policy in China, of which 22 are vegetable pilot counties. Technical training and organic fertilizer subsidies are two main measures. Agricultural technology extension departments provide training on the role of organic fertilizer on soil and composting technology, and give technical guidance to large-scale growers, cooperatives and other agricultural operators. Each county receives a national fund subsidy of 10 million yuan and a certain percentage of subsidy funding provided by local county government.

In this paper, we hypothesize that when making behavioural decisions, farmers are rational in line with the principle of optimization. In a market economy, farmers' decisions on using fertilizer are   characterized not only by their concern about the returns on different technologies, but also by their risk-averse nature that they are more likely to adopt easily available fertilizer in the absence of policy incentives. At the same time, farmers are social people and may be influenced by external policies. Policy intervention is an effective means to solve the internalization of agricultural production technology externalities, which can change the constraints of farmers' decision-making and is a strong determinant of farmers' adoption in new technologies. So, how do policy measures of training and subsidies influence farmers' behaviour? What role do farmers' psychological cognition play in policy measures and farmers' behaviour? This paper applies the Theory of Planned Behaviour (TPB) to explain the effects of two policy measures on farmers' behaviour and assesses the mediating effect of farmers' psychological cognition between two measures and OSCF adoption to reveal the external and internal causes of farmers’ behaviour.  

The influence of farmers' cognition on the effectiveness of environmentally friendly agriculture policy implementation has received wide attention. Some studies have concluded that the cognition that OSCF can improve prices of apples, and the cognition about ease of organic fertilizer has significant positive effects on fruit farmers' organic fertilizer use behaviour [9]. Moreover, some studies have confirmed that farmers have low cognition of environmental pollution from chemical fertilizer, insufficient knowledge in fertilizer application and reduction policies, and low willingness to reduce application [10]. Farmers' environmental friendly cognition of organic fertilizer cannot promote farmers' organic fertilizer use [11]. But some studies have verified that ecological cognition partially mediated the relationship between social norms and farmers' willingness to recycle mulch by using ecological cognition [12].

This paper aims to contribute to the existing literature by further validating the role of farmers' cognition in agri-environmental policies and farmers' behaviour, and clarifying the variability of farmers’ cognition and the path of mediating effects. First, we construct a systematic research framework, and analyse the external policy environment with policy measures and the interactions between farmers' cognition and behaviour. Second, we analyse the interaction among agricultural environmental policies, farmers' cognition and behaviour and considering the mediating effects on farmers' awareness of economic value, resource capacity and ecosystem impact. Finally, we consider the path of mediating effects of farmers' cognition, and examine the role of farmers' cognition in the share of these mediating effects between agri-environmental policies and farmers' behaviour. Moreover, we measure the share of mediating effects in farmers' cognition under two policy measures including technical training and organic fertilizer subsidies.

The paper comprises five sections. Section 2 establishes the theoretical framework. Section 3 describes the research data collection, variable selection of farmers' main characteristics, and modelling methods. The results are reported in part 4. Finally, Section 5 provides conclusions and the discussion of policy recommendations."

Point 4: The empirical survey is insufficient to provide an answer to the research question.1. It is unclear how much training the farmers received.2. It is unclear how high the subsidies are and whether they compensate for the additional costs.3. It is unclear in what form organic fertilizer is applied and whether it is available.4. Why is the distinction between sustainable and environment?5 . Only school education is asked, but not practical experience.6. Why the distinction between sustainable and environmental in relation to soil? 7. Only the farm size is collected, what about the greenhouse area?

Response 4: For the variable selection aspect, we have restated it. The relevant variables in Table 2 were also modified. The details are as follows.

"The entities of implementation are organic fertilizer enterprises and cooperatives. 1) Organic fertilizer enterprises in the region collect livestock and poultry manure and produce organic fertilizer to vegetable farmers. Farmers can purchase organic fertilizer with lower prices than market prices and organic fertilizer enterprises also can get subsides. 2) Cooperatives with subsidies are encouraged to carry out composting and fermentation of livestock and poultry manure in the vicinity, and members can receive more favourable prices of organic fertilizer and technical guidance. There are two ways to send organic fertilizer subsidies. One is that farmers who make fertilizer by themselves or purchase compost in the pilot regions will receive a standard subsidy of 300 yuan per ton. Second is that farmers using commercial organic fertilizer will subsided with 50% of the price on organic fertilizer. The percentage of receiving subsidies is 29%, which was low. In Pingyuan County, organic fertilizer production mainly relies on enterprises processing commercial organic fertilizer, and farmers get preferential prices for organic fertilizer at an average price of 2000 yuan/t, while the average price of commercial organic fertilizer in Yucheng County, a neighbouring non-policy area county, is 2500 yuan/t. The price of commercial organic fertilizer in Pingyuan County is 25% better. In Anqiu County, organic fertilizer is mainly composted by cooperatives, according to the research sample data, it can be seen that the cost of compost after the subsidy is about 200 yuan/t, some vegetable farmers can also get free compost, while the average price of compost in Qingzhou City is about 440 yuan/t. After getting the subsidy, the cost of using compost has dropped at least 50%. In the survey, Farmers are indirect beneficiaries of the subsidy, so they only perceive the reduction of price in organic fertilizer and do not know the exact amount of the subsidy. So we set it as 0 or 1 variable, and 1 represents farmers got financial subsidies.

Based on CBT in Part 2, the cognition of economic value, resource capacity and ecological impact may have a mediating effect on farmers' adoption of organic fertilizer as a substitute for chemical fertilizer. We used a 5-point scale for these three variables, with 1 indicating strongly disagree and 5 indicating strongly agree (Table 1).

The control variables are vegetable grower household head characteristics, household characteristics and vegetable plot characteristics, mainly to see whether there are other factors influencing vegetable farmers' behaviour of using organic fertilizer instead of chemical fertilizer."

Point 5: Figure.3 left not necessary, results are already included in Tab 1.“Table 2. Regional differences of farmers OSCF” is interesting, but it lacks an attempt to explain the large differences between regions. Line 293: “As rational smallholder farmers, vegetable farmers focus on maximizing their economic profits, and the cost of using organic fertilizers is higher than that of chemical fertilizers.” Yes, that is economic rationality. But there is a lack of information in the text and in the survey about the costs and government subsidies or the relationship between them. Line 323. “Thus, hypothesis H3 has been verified” Surprisingly how can a hypothesis be verified that has not been formed (There is no H3 in your Text!)Line 363: “As a kind of ecological protection compensation program, the organic fertilizer subsidy program makes up the opportunity cost of organic fertilizer loss for green-364 house vegetable farmers”. How can this be deduced from the empirical findings in this paper? The paper should be more clearly focused and only those aspects should be discussed that were also empirically queried. However, it is questionable whether this captures and explains reality.

Response 5: For the study results, we further revised and improved them. Taking into account the structure of article, we have adjusted the original regional analysis from the results to the discussion section for region was not a key consideration in our original hypothesis. In our analysis of study, we found significant regional differences in OSCF adoption behaviour among vegetable farmers and this was closely related to the subsidy policy, the focus of our study. We believe that the analysis of regional differences should be placed in the discussion section, so that we can further analyse the relationship between regions, subsidy policies and OSCF adoption behaviour. So we add data to highlight the differences in regions and further analyzed the relationship between the cost of organic fertilizer and subsidy. The details are as follows.

"We found significant variability in the adoption of OSCF by greenhouse vegetable growers in four counties studied. First, we selected two policy counties and two non-policy counties in the study area selection. For the non-policy areas, some training was adopted with no subsidy, and the adoption rate of OSCF among vegetable farmers was low; the pilot counties had policy support and the adoption rate was higher. Secondly, two policy counties had differences in organic fertilizer production methods. Pingyuan county mainly relied on agricultural enterprises to produce commercial organic fertilizer. Although they get government subsidy, the sales price of organic fertilizer was only 20% better than the market price (Table 6). So, the adoption rate of vegetable farmers did not reach a very high level. On the contrary, Anqiu County mainly adopted the cooperative composting and fermentation in the vicinity with lower production cost of organic fertilizer. Vegetable farmers can enjoy lower price of organic fertilizer, which is 54.5% better than the market price (Table 6). According to the differences in regions, each area should choose appropriate ways to promote OSCF. "

Table 6 Investigated counties and vegetable farm household samples

Region

Adoption ratio (%)

Percentage of farmers receiving subsidies

(%)

Price of organic fertilizer

(yuan/t)

Method of getting organic fertilizer

(1=commercial organic fertilizer, 2=compost)

Non-policy county

Yucheng

29.5

0

2500

1

Qingzhou

39.2

0

440

2

Policy county

Pingyuan

51.1

50

2000

1

Anqiu

86.7

77.7

200

2

Note: Data came from the authors’ survey

Point6: Missing references: XIE Hai-kuan and XU Chi, 2019; 68 Yanli, 2015Lu et 79 al., 2019;Schultz, 1964

Response 6: We've added the relevant missing references, thanks very much!

Author Response

Dear Reviewer:

Many thanks for your detailed and valuable comments and suggestions on our manuscript entitled “What motivates greenhouse vegetable farmers to adapt Organic-Substitute-Chemical-Fertilizer(OSCF)? An empirical study from Shandong, China?” (ijerph-2117168). The comments and suggestions are very helpful for improving our paper. We have made revision based on the comments and suggestions. Please find our response as follows.

Point1:It would be interesting to include a map of China to locate the study area.

Response1:We modified the figure to show the location of Shandong in China.

Point2: Significant differences appear in the rate of adoption by farmers among the 4 selected areas. What are these differences due to? The authors should include and discuss it.

Response2: Thank you for your suggestion. For the study results, we further revised and improved them. Taking into account the structure of article, we have adjusted the original regional analysis from the results to the discussion section for region was not a key consideration in our original hypothesis. In our analysis of study, we found significant regional differences in OSCF adoption behaviour among vegetable farmers and this was closely related to the subsidy policy, the focus of our study. We believe that the analysis of regional differences should be placed in the discussion section, so that we can further analyse the relationship between regions, subsidy policies and OSCF adoption behaviour. So we add data to highlight the differences in regions and further analyzed the relationship between the cost of organic fertilizer and subsidy. The details are as follows.

"We extracted the sample counties which are policy support areas and general areas. Pingyuan County and Anqiu County are the first batch of policy areas of organic fertilizer substitution for chemical fertilizer on greenhouse vegetables and will receive corresponding policy support. For comparison, we chose two counties with more vegetable cultivation next to these two counties, as Yucheng County and Qingzhou County.

We found significant variability in the adoption of OSCF by greenhouse vegetable growers in four counties studied. First, we selected two policy counties and two non-policy counties in the study area selection. For the non-policy areas, some training was adopted with no subsidy, and the adoption rate of OSCF among vegetable farmers was low; the pilot counties had policy support and the adoption rate was higher. Secondly, two policy counties had differences in organic fertilizer production methods. Pingyuan county mainly relied on agricultural enterprises to produce commercial organic fertilizer. Although they get government subsidy, the sales price of organic fertilizer was only 20% better than the market price (Table 6). So, the adoption rate of vegetable farmers did not reach a very high level. On the contrary, Anqiu County mainly adopted the cooperative composting and fermentation in the vicinity with lower production cost of organic fertilizer. Vegetable farmers can enjoy lower price of organic fertilizer, which is 54.5% better than the market price (Table 6). According to the differences in regions, each area should choose appropriate ways to promote OSCF. "

Table 6 Investigated counties and vegetable farm household samples

Region

Adoption ratio (%)

Percentage of farmers receiving subsidies

(%)

Price of organic fertilizer

(yuan/t)

Method of getting organic fertilizer

(1=commercial organic fertilizer, 2=compost)

Non-policy county

Yucheng

29.5

0

2500

1

Qingzhou

39.2

0

440

2

Policy county

Pingyuan

51.1

50

2000

1

Anqiu

86.7

77.7

200

2

Note: Data came from the authors’ survey

Point3:In general, as the article is written, the presentation of the data in tables and figures and the subsequent discussion and comments of the results is not adequate and its understanding and follow-up is difficult. This part should be rewritten to be more clear and direct.

Response3: Thank you very much for your suggestion. We have rewritten the data presentation of graphs and the discussion of study results. The details are as follows.

3.2 Variable selections

Based on the above theoretical framework, the dependent variable is whether to adopt OSCF. The assigned value is 1 for the adoption of OSCF, and 0 for non-adoption. The independent variables include three types: 1) Core independent variables, including technical training and financial subsidies. 2) Mediate variables, including farmers' cognition of economic value, resource capacity, and ecosystem impact of OSCF. 3) Control variables, including the characteristics of the head of household (e.g. age and education level), household characteristics (e.g. cooperative relationship, labour force numbers and greenhouse vegetable area) and characteristics of greenhouse vegetable plots (e.g. soil fertility and property rights’ characteristics). The description of these variables is provided in Table 1.

The adoption rate of OSCF reached 52% and farmers can obtain two types of organic fertilizer in the regions. One is commercial organic fertilizer, which is processed by organic fertilizer enterprises with lower use costs, smaller environmental pollution, but relatively higher price about between 2 yuan/kg – 4 yuan/kg. The other category is composting fertilizer known as manure, which achieves resource utilization through composting fermentation of livestock and poultry manure.

The reason of taking technical training and financial subsidies as core independent variables is that these two measures are main ways to promote the policy of OSCF. Each county formulates corresponding promotion measures based on the amount of organic fertilizer resources and the area of greenhouse vegetable. On the one hand, counties may carry out promotional measures, such as strengthening chemical fertilizer reduction publicity and training. In terms of training, 46% of the farmers receive OSCF-related training. The method of training includes intensive study courses and field training. The intensive study course mainly focuses on theoretical study of fertilisation content, and field training mainly involves hiring experts to provide on-site guidance on practical problems. The training includes the amount, ratio and timing of fertiliser application for different vegetables. During the questionnaire survey, farmers participated in a wide range of training categories, such as vegetable storage and pesticide spraying. So, they could only remember whether they had attended OSCF training and could not accurately state the number of times they had attended OSCF training. Therefore, this paper use whether they had attended training as an explanatory variable.

On the other hand, each county supported by the policy will receive subsidy of 10 million yuan and get matching funds according to its financial level. The entities of implementation are organic fertilizer enterprises and cooperatives. 1) Organic fertilizer enterprises in the region collect livestock and poultry manure and produce organic fertilizer to vegetable farmers. Farmers can purchase organic fertilizer with lower prices than market prices and organic fertilizer enterprises also can get subsides. 2) Cooperatives with subsidies are encouraged to carry out composting and fermentation of livestock and poultry manure in the vicinity, and members can receive more favourable prices of organic fertilizer and technical guidance. There are two ways to send organic fertilizer subsidies. One is that farmers who make fertilizer by themselves or purchase compost in the pilot regions will receive a standard subsidy of 300 yuan per ton. Second is that farmers using commercial organic fertilizer will subsided with 50% of the price on organic fertilizer. The percentage of receiving subsidies is 29%, which was low. In Pingyuan County, organic fertilizer production mainly relies on enterprises processing commercial organic fertilizer, and farmers get preferential prices for organic fertilizer at an average price of 2000 yuan/t, while the average price of commercial organic fertilizer in Yucheng County, a neighbouring non-policy area county, is 2500 yuan/t. The price of commercial organic fertilizer in Pingyuan County is 25% better. In Anqiu County, organic fertilizer is mainly composted by cooperatives, according to the research sample data, it can be seen that the cost of compost after the subsidy is about 200 yuan/t, some vegetable farmers can also get free compost, while the average price of compost in Qingzhou City is about 440 yuan/t. After getting the subsidy, the cost of using compost has dropped at least 50%. In the survey, Farmers are indirect beneficiaries of the subsidy, so they only perceive the reduction of price in organic fertilizer and do not know the exact amount of the subsidy. So we set it as 0 or 1 variable, and 1 represents farmers got financial subsidies.

Based on CBT in Part 2, the cognition of economic value, resource capacity and ecological impact may have a mediating effect on farmers' adoption of organic fertilizer as a substitute for chemical fertilizer. We used a 5-point scale for these three variables, with 1 indicating strongly disagree and 5 indicating strongly agree (Table 1).

The control variables are vegetable grower household head characteristics, household characteristics and vegetable plot characteristics, mainly to see whether there are other factors influencing vegetable farmers' behaviour of using organic fertilizer instead of chemical fertilizer.

Table 1 Variables and descriptive statistics

Type of variables

Description and assignment

Maximum

Minimum

Mean

SD

Dependent variables

whether to adopt OSCF

1=Yesï¼›0=No

1

0

0.53

0.49

Core independent variables

Training(X1)

1=Yesï¼›0=No

1

0

0.45

0.49

Subsidy(X2)

1=Yesï¼›0=No

1

0

0.29

0.45

Mediate variables

Economic value(M1)

Do you think OSCF will increase vegetable income? Level 5:1= strongly disagree, 5= strongly agree

5

1

3.25

1.07

Resource capacity(M2)

Do you think OSCF is good for resource capacity of vegetable plots? Level 5: 1= strongly disagree, 5= strongly agree

5

1

3.03

1.18

Ecosystem impact(M3)

Do you think OSCF can improve agri-ecological environment? Level 5: 1= strongly disagree, 5= strongly agree

5

1

3.11

0.99

Control variables

The head of the household characteristics

Age(C1)

ordinal

83

27

52.02

8.58

Education(C2)

1= primary school and illiterate, 2= junior school, 3= high school, 4= junior college and higher education

4

1

2.00

0.65

Family characteristics

Cooperative(C3)

Whether to join the cooperative? 1=Yesï¼›0=No

1

0

0.47

0.50

Labour force(C4)

ordinal

6

0

2.87

1.00

Greenhouse vegetable area(C5)

Ha

5.09

0.03

0.51

0.43

 Characteristics of greenhouse vegetable plots

Soil fertility(C6)

1= barren; 2 = in general; 3 = fertile

3

1

2.53

0.54

Property rights(C7)

1= management right, 2= Self-contracting rights

1

0

0.49

0.50

Point4: For example: Figure 3 and the information it contains does not seem to be commented and discussed in the text. It must be reviewed. The number of the figure should be cited in the text for reference.

Response4:Figure 3 has been deleted, and we have modified it by using a table format (Table 6). The relevant content is in section 5.3.

Point5:For example line 385: it would be convenient to indicate Table 5 after 63.58% and 48.57%.

Response5:Thank you very much, we have marked it.

Reviewer 3 Report

This is a very intresting research work considering an issue which is not frequently examined in the greenhouse sector. Some points needs to be clarified to enhnace its' quality

1. Please add a paragraph where you will mention the actual availiability of organig fertilizers. The thoereticall and sum approach is  of course accepted ,however, do we have a clear vision if the organic fertilizers are available easily in every location in China for example (or other countries worldwide). Please explain.

2. In Table 2, what's the exaplanation for such differences betwen regions (are the parameters that you explain afterwards connected with the regions?). I. e in X region the farmers are well trained, or subsidies are more beneficiar in region Y -if there are differences between regions in China?). Please add in results and conclusion session a paragraph to expalin regional differnce 

3. Minor language improvment in Paragraphs 5.2 and 5.3.

Author Response

Dear Reviewer:

Many thanks for your detailed and valuable comments and suggestions on our manuscript entitled “What motivates greenhouse vegetable farmers to adapt Organic-Substitute-Chemical-Fertilizer(OSCF)? An empirical study from Shandong, China?” (ijerph-2117168). The comments and suggestions are very helpful for improving our paper. We have made revision based on the comments and suggestions. Please find our response as follows.

Point1: Please add a paragraph where you will mention the actual availability of organic fertilizers. The theoretical and sum approach is of course accepted ,however, do we have a clear vision if the organic fertilizers are available easily in every location in China for example (or other countries worldwide). Please explain.

Response1: We add a note on the availability of organic fertilizers in section 3.2 Variable selections. The details are as follows.

3.2 Variable selections

The entities of implementation are organic fertilizer enterprises and cooperatives. 1) Organic fertilizer enterprises in the region collect livestock and poultry manure and produce organic fertilizer to vegetable farmers. Farmers can purchase organic fertilizer with lower prices than market prices and organic fertilizer enterprises also can get subsides. 2) Cooperatives with subsidies are encouraged to carry out composting and fermentation of livestock and poultry manure in the vicinity, and members can receive more favourable prices of organic fertilizer and technical guidance. There are two ways to send organic fertilizer subsidies. One is that farmers who make fertilizer by themselves or purchase compost in the pilot regions will receive a standard subsidy of 300 yuan per ton. Second is that farmers using commercial organic fertilizer will subsided with 50% of the price on organic fertilizer. The percentage of receiving subsidies is 29%, which was low. In Pingyuan County, organic fertilizer production mainly relies on enterprises processing commercial organic fertilizer, and farmers get preferential prices for organic fertilizer at an average price of 2000 yuan/t, while the average price of commercial organic fertilizer in Yucheng County, a neighbouring non-policy area county, is 2500 yuan/t. The price of commercial organic fertilizer in Pingyuan County is 25% better. In Anqiu County, organic fertilizer is mainly composted by cooperatives, according to the research sample data, it can be seen that the cost of compost after the subsidy is about 200 yuan/t, some vegetable farmers can also get free compost, while the average price of compost in Qingzhou City is about 440 yuan/t. After getting the subsidy, the cost of using compost has dropped at least 50%. In the survey, Farmers are indirect beneficiaries of the subsidy, so they only perceive the reduction of price in organic fertilizer and do not know the exact amount of the subsidy. So we set it as 0 or 1 variable, and 1 represents farmers got financial subsidies.

Point2: In Table 2, what's the explanation for such differences between regions (are the parameters that you explain afterwards connected with the regions?). I. e in X region the farmers are well trained, or subsidies are more beneficial in region Y -if there are differences between regions in China?). Please add in results and conclusion session a paragraph to explain regional difference.

Response2:

For the study results, we further revised and improved them. Taking into account the structure of article, we have adjusted the original regional analysis from the results to the discussion section for region was not a key consideration in our original hypothesis. In our analysis of study, we found significant regional differences in OSCF adoption behaviour among vegetable farmers and this was closely related to the subsidy policy, the focus of our study. We believe that the analysis of regional differences should be placed in the discussion section, so that we can further analyse the relationship between regions, subsidy policies and OSCF adoption behaviour. So we add data to highlight the differences in regions and further analyzed the relationship between the cost of organic fertilizer and subsidy. The details are as follows.

"We found significant variability in the adoption of OSCF by greenhouse vegetable growers in four counties studied. First, we selected two policy counties and two non-policy counties in the study area selection. For the non-policy areas, some training was adopted with no subsidy, and the adoption rate of OSCF among vegetable farmers was low; the pilot counties had policy support and the adoption rate was higher. Secondly, two policy counties had differences in organic fertilizer production methods. Pingyuan county mainly relied on agricultural enterprises to produce commercial organic fertilizer. Although they get government subsidy, the sales price of organic fertilizer was only 20% better than the market price (Table 6). So, the adoption rate of vegetable farmers did not reach a very high level. On the contrary, Anqiu County mainly adopted the cooperative composting and fermentation in the vicinity with lower production cost of organic fertilizer. Vegetable farmers can enjoy lower price of organic fertilizer, which is 54.5% better than the market price (Table 6). According to the differences in regions, each area should choose appropriate ways to promote OSCF. "

Table 6 Investigated counties and vegetable farm household samples

Region

Adoption ratio (%)

Percentage of farmers receiving subsidies

(%)

Price of organic fertilizer

(yuan/t)

Method of getting organic fertilizer

(1=commercial organic fertilizer, 2=compost)

Non-policy county

Yucheng

29.5

0

2500

1

Qingzhou

39.2

0

440

2

Policy county

Pingyuan

51.1

50

2000

1

Anqiu

86.7

77.7

200

2

Note: Data came from the authors’ survey

Point3: Minor language improvement in Paragraphs 5.2 and 5.3

Response3:Thank you very much for your suggestion. We have corrected and upgraded the language in 5.2 and 5.3.

Reviewer 4 Report

 I herein elaborate on my concerns below and hope they help authors strengthen their manuscripts on this timely and relevant topic.

1.      In Table 1 wherein the descriptive statistics regarding the variables of the paper are presented, the control variable “Education” is classified into 4 categories each representing varying degrees of educational attainment. Is this way of classifying educational attainment based on the authors’ contention? Or is it based on a standard? It shall be clarified [215-226].

2.      There exists no explanation as to the Figure that follows Table 2 [275-276]. As the Figure is related to the main independent variables and mediators, the authors are advised to both number and elaborate on the Figure.

3.      The information as to whether statistically significant differences regarding select control variables exist amongst sampled farmers is lacking. Its addition to the text would be quite informative for readers.

4.      As expected, in Table 3, organic fertilizer subsidies positively affect the farmer’s behavior to utilize ‘OSCF’, validating the authors’ first hypothesis. Does this particular result have support in the extant literature? In addition, farmers’ age, educational attainment, and agricultural cooperative membership status have no significant impact on their decision to adopt OSCF. How do authors explain that? Especially estimated negative coefficient as to the variable ‘Education’ [278-307].

5.      Based on the findings as to the estimated coefficients of select variables, the authors concluded that Hypothesis 2 was validated. I failed to see the evidence with regard to the authors’ confirmation in Table 3. And Table 3 surely does not test hypothesis 2. It shall be corrected. [278-307].

6.      Table 4 represents the results of the effects of mediators on farmers’ adaption of OSCF. Evidently, the results evidence the existence of mediator relations. Thus, the authors concluded that “….hypothesis H3 has been verified.”. Hypothesis 3 does not exist as far as I see. There are two hypotheses that are subject to testing, Hypothesis 1 and Hypothesis 2. The authors must have mistaken Hypothesis 3 for Hypothesis 2. It shall be corrected. [308-326]

7.      It would be more appropriate if it were explained how 318 observation selection was determined and which methodology is used for it.

8.      The authors have combined behavioral analysis with survey data but have failed to provide justifications for their sample size adequacy. Considering the population of select counties under Shandong Province out of which their sample is formed and given the authors’ sample size of 318 respondents, Is it enough to make meaningful conclusions based on the estimation results?[1][187-214]

[1] The authors might want to check: Penne, M. A., & Levy, P. S. (2014). Sample Size Adequacy in Surveys. Wiley StatsRef: Statistics Reference Online. doi:10.1002/9781118445112.stat057.

Author Response

Dear Reviewer:

Many thanks for your detailed and valuable comments and suggestions on our manuscript entitled “What motivates greenhouse vegetable farmers to adapt Organic-Substitute-Chemical-Fertilizer(OSCF)? An empirical study from Shandong, China?” (ijerph-2117168). The comments and suggestions are very helpful for improving our paper. We have made revision based on the comments and suggestions. Please find our response as follows.

Point1: In Table 1 where in the descriptive statistics regarding the variables of the paper are presented, the control variable “Education” is classified into 4 categories each representing varying degrees of educational attainment. Is this way of classifying educational attainment based on the authors’ contention? Or is it based on a standard? It shall be clarified [215-226].

Response1: We revise the description of education in Table 1. The division of education into four categories is based on the stage of education in China and the level of education generally received by farmers. For example, this paper also classifies the educational level of the survey respondents according to the educational stage of China [1]. (1.Lu, H.; Zhang, P.; Hu, H.; Xie, H.; Yu, Z.; Chen, S. Effect of the Grain-Growing Purpose and Farm Size on the Ability of Stable Land Property Rights to Encourage Farmers to Apply Organic Fertilizers. Journal of Environmental Management 2019, 251, 109621, doi:10.1016/j.jenvman.2019.109621.)

Type of variables

Description and assignment

Maximum

Minimum

Mean

SD

Education(C2)

1= primary school and illiterate, 2= junior school, 3= high school, 4= college and higher education

4

1

2.00

0.65

Point2: here exists no explanation as to the Figure that follows Table 2 [275-276]. As the Figure is related to the main independent variables and mediators, the authors are advised to both number and elaborate on the Figure.

Response2: Figure 3 has been deleted, and we have modified it by using a table format (Table 6). The relevant content is in section 5.3. For the study results, we further revised and improved them. Taking into account the structure of article, we have adjusted the original regional analysis from the results to the discussion section for region was not a key consideration in our original hypothesis. In our analysis of study, we found significant regional differences in OSCF adoption behaviour among vegetable farmers and this was closely related to the subsidy policy, the focus of our study. We believe that the analysis of regional differences should be placed in the discussion section, so that we can further analyse the relationship between regions, subsidy policies and OSCF adoption behaviour. So we add data to highlight the differences in regions and further analyzed the relationship between the cost of organic fertilizer and subsidy. The details are as follows.

"We found significant variability in the adoption of OSCF by greenhouse vegetable growers in four counties studied. First, we selected two policy counties and two non-policy counties in the study area selection. For the non-policy areas, some training was adopted with no subsidy, and the adoption rate of OSCF among vegetable farmers was low; the pilot counties had policy support and the adoption rate was higher. Secondly, two policy counties had differences in organic fertilizer production methods. Pingyuan county mainly relied on agricultural enterprises to produce commercial organic fertilizer. Although they get government subsidy, the sales price of organic fertilizer was only 20% better than the market price (Table 6). So, the adoption rate of vegetable farmers did not reach a very high level. On the contrary, Anqiu County mainly adopted the cooperative composting and fermentation in the vicinity with lower production cost of organic fertilizer. Vegetable farmers can enjoy lower price of organic fertilizer, which is 54.5% better than the market price (Table 6). According to the differences in regions, each area should choose appropriate ways to promote OSCF. "

Table 6 Investigated counties and vegetable farm household samples

Region

Adoption ratio (%)

Percentage of farmers receiving subsidies

(%)

Price of organic fertilizer

(yuan/t)

Method of getting organic fertilizer

(1=commercial organic fertilizer, 2=compost)

Non-policy county

Yucheng

29.5

0

2500

1

Qingzhou

39.2

0

440

2

Policy county

Pingyuan

51.1

50

2000

1

Anqiu

86.7

77.7

200

2

Note: Data came from the authors’ survey

Point3: The information as to whether statistically significant differences regarding select control variables exist among sampled farmers is lacking. Its addition to the text would be quite informative for readers

Response3: We have revised the expressions that are inappropriate to the text. Since the control variables are not our main concern, we only briefly express whether there is a significant effect of control variables and do not elaborate on exactly how much a significant difference is.

Point4: As expected, in Table 3, organic fertilizer subsidies positively affect the farmer’s behavior to utilize ‘OSCF’, validating the authors’ first hypothesis. Does this particular result have support in the extant literature? In addition, farmers’ age, educational attainment, and agricultural cooperative membership status have no significant impact on their decision to adopt OSCF. How do authors explain that? Especially estimated negative coefficient as to the variable ‘Education’ [278-307].

Response4: In this paper, Table 2 also shows that there is no significant effect of farmers' age and education on farmers' adoption of organic fertilizer and the estimated coefficient of "education" is negative. Possible explanation: farmers with lower education have weaker personal capital and are more likely to have a near-zero opportunity cost of time that they may be able to spend more time on collecting manure to make farmyard manure and apply organic fertilizer [2]. However, although insignificant variables are generally not discussed in detail, the explanation can also be currently inferred from our actual research and existing theories and a more detailed analysis needs further discussion. ( 2. Li, B.; Zeng, Q. The Effect of Land Right Stability on the Application of Fertilizer Reduction Technologies—Evidence from Large-Scale Farmers in China. Sustainability 2022, 14, 8059, doi:10.3390/su14138059.)

Point5: Based on the findings as to the estimated coefficients of select variables, the authors concluded that Hypothesis 2 was validated. I failed to see the evidence with regard to the authors’ confirmation in Table 3. And Table 3 surely does not test hypothesis 2. It shall be corrected. [278-307].

Response5: Thank you for your reminder, we have corrected the error.

Point6: Table 4 represents the results of the effects of mediators on farmers’ adaption of OSCF. Evidently, the results evidence the existence of mediator relations. Thus, the authors concluded that “….hypothesis H3 has been verified.”. Hypothesis 3 does not exist as far as I see. There are two hypotheses that are subject to testing, Hypothesis 1 and Hypothesis 2. The authors must have mistaken Hypothesis 3 for Hypothesis 2. It shall be corrected. [308-326]

Response6: Thank you for your reminder, we have corrected the error.

Point7: It would be more appropriate if it were explained how 318 observation selection was determined and which methodology is used for it.

Response7: In the selection of sample counties, we extracted the sample counties which are policy support areas and general areas. Pingyuan County and Anqiu County are the first batch of policy areas of organic fertilizer substitution for chemical fertilizer on greenhouse vegetables and will receive corresponding policy support. For comparison, we chose two counties with more vegetable cultivation next to these two counties, as Yucheng County and Qingzhou County. The sample counties are shown in Fig.2.

Point8: The authors have combined behavioral analysis with survey data but have failed to provide justifications for their sample size adequacy. Considering the population of select counties under Shandong Province out of which their sample is formed and given the authors’ sample size of 318 respondents, Is it enough to make meaningful conclusions based on the estimation results?[1][187-214]

Response8: Shandong is the main production province of "vegetable basket" products in north China. The area of greenhouse vegetables in Shandong province is about 14 million mu accounting for about 1/4 of country's total area, and the output reaches more than 50 million tons per year. It has become the main hub of Chinese vegetable industry, and their products are sold to major domestic vegetable markets. Therefore, selecting Shandong as studying area to analyse OSCF adoption behaviour of vegetable farmers ensures the quantity and quality of the sample. We randomly selected 5 villages from each of the four counties. 20 households from each village were randomly selected according to the list provided by the village committee. Eventually a total of 318 valid questionnaires were obtained. Among this sample of 318, both policy support areas and areas without policy intervention were covered. Areas with subsidized policies were selected as intervention group, and areas without subsidized policy support were selected for investigation in nearby counties with similar situations. So the analysis of this section using a sample of 318 can lead to meaningful conclusions.
